# Patterns of pollen and resource limitation of fruit production in *Vaccinium uliginosum* and *V. vitis-idaea* in Interior Alaska

**Lindsey Viann Parkinson**[ORCID]\*, **Christa P. H. Mulder**

Institute of Arctic Biology and Department of Biology and Wildlife, University of Alaska Fairbanks, Fairbanks, Alaska, United States of America

\* parkinson.lindsey@protonmail.com

**Data Availability Statement:** All relevant data are within the manuscript and its Supporting Information files.

## Abstract

Many recent studies assessing fruit productivity of plants in the boreal forest focus on inter-annual variability across a forested region, rather than on environmental variability within the forest. Frequency and severity of wildfires in the boreal forest affect soil moisture, canopy, and community structure at the landscape level, all of which may influence overall fruit production at a site directly or indirectly. We evaluated how fruit production in two boreal shrubs, *Vaccinium uliginosum* (blueberry) and *V. vitis-idaea* (lingonberry), was explained by factors associated with resource availability (such as canopy cover and soil conditions) and pollen limitation (such as floral resources for pollinators and pollen deposition) across boreal forest sites of Interior Alaska in 2017. We classified our study sites into upland and lowland sites, which differed in elevation, soil moisture, and active layer. We found that resource and pollen limitation differed between the two species and between uplands and lowlands. Lingonberry was more pollen limited than blueberry, and plants in lowland sites were more pollen limited relative to other sites while plants in upland sites were relatively more resource limited. Additionally, canopy cover had a significant negative effect in upland sites on a ramet's investment in reproductive tissues and leaves versus structural growth, but little effect in lowland sites. These results point to importance of including pollinator service as well as resource availability in predictions for changes in berry abundance.

## Introduction

At least 50 species of plants produce fleshy fruits (hereafter: "berries") in Alaska [1]. In Interior Alaska, a region bordered by the Alaska Range to the south and the Brooks Range to the north, *Vaccinium vitis-idaea* L. (lingonberry) and *V. uliginosum* L. (lowbush blueberry, hereafter: blueberry) are two of the fruits most commonly consumed by both humans and animals [2]. Many species including bears (*Ursus arctos* and *U. americanus*), foxes (*Vulpes vulpes)*, and voles (e.g., *Myodes rutilus*) eat the berries [3–6]. Nearly three quarters of all berries collected in rural communities in Interior Alaska in 2015 were from these two species [7]. Berry production is a multi-year process dependent upon weather, pollinator activity, light availability, and

**Funding:** Funding was provided by the Bonanza Creek LTER (NSF grant no. DEB-1636476). The Bonanza Creek LTER is a member of the U.S. LTER Network which is supported by the National Science Foundation and the USDA Forest Service, Pacific Northwest Research Station. The funders had no role in study design, data collection and analysis, decision to publish, or preparation of the manuscript.

**Competing interests:** The authors have declared that no competing interests exist.

soil conditions [8,9]. Recent studies assessing berry production in boreal plants have focused on interannual variability across a region [10,11] but berry production varies within the region as well [12,13]. Due to the multi-year development period of *Vaccinium* flowers, interannual models of fruit production account for some of the effects of changing weather and climate; however, abiotic factors that affect local growth patterns are often overlooked.

Interior Alaska is undergoing rapid climate change, altering not just temperature but also the frequency, severity, and extent of wildfire [14–16]. Understanding how *Vaccinium* berry production responds to heterogeneous environmental factors such as variation in resource availability (limiting growth and berry development) and variation in pollinator service (limiting compatible pollen deposition and subsequent fertilization) within Alaska's boreal forest can provide a foundation for modelling berry crops for humans and animals. Models assessing how changes in wildfire, soil moisture, and permafrost in Interior Alaska may affect plant community structure, already exist [17–19]. However, vegetative plant growth, fruit, and seed availability are not always correlated with one another [20]. Fruit production in all flowering plants is limited by four factors: 1) resources (e.g., light and soil moisture), 2) pollination, 3) external factors such as herbivory, disease, or harsh weather, and 4) genetics [21–23]. Here we focus on resource and pollen limitation.

Slope, aspect, elevation, and fire frequency drive plant community structure in the boreal zone [24–26]. North-facing slopes receive limited sunlight, have cold, poorly drained soils underlain with permafrost, and are primarily composed of *Picea mariana* (black spruce) stands with a moss understory [24, 26]. South facing slopes tend to have warmer, well drained soils occupied by deciduous trees and *P. glauca* (white spruce) [24,25]. Slope, aspect, and elevation have not changed over the past century, but wildfires are getting larger and returning more quickly across North America's boreal forest regions [14, 27,28]. Fire shapes ecosystem dynamics including plant succession and soil condition. In most situations, low shrubs are the dominant cover for 10–20 years after a fire, after which tall shrubs and deciduous trees begin to take over and the canopy closes, limiting the light available [29]. If the seed bank survived the wildfire, deciduous trees generally give way to spruce and the canopy opens again [24,25].

Black spruce forests on north facing and lowland sites are typically underlain by permafrost, which leads to cold, wet soils with low nutrient availability [26]. The presence of shallow permafrost cools the soil and inhibits drainage, water collects from weather events through the growing season as well as from the thawing ground. Fires can remove much of the moss and soil layer that insulates the permafrost, drastically increasing the depth of the active layer (the layer that freezes and thaws annually) depth and thus moisture and temperature conditions of the forest stand [15]. We would therefore expect fire history to affect resource availability both by altering the canopy cover and by altering soil moisture and depth of thaw as has been seen in other northern regions such as Fennoscandia [30].

While they are closely related, blueberries and lingonberries differ in their life history strategy. Lingonberries produce thick, evergreen leaves that last about three years (CPH Mulder, pers. obs.) and replace 39% of standing biomass each year, while blueberries produce deciduous leaves and have an annual turnover of 62% standing biomass [31]. Blueberries thus fall closer to the resource acquisitive end of the leaf economic spectrum [32,33] and are potentially able to respond to changes in habitat conditions more quickly than lingonberries, which are on the resource conservative end of the spectrum. We would therefore expect a stronger relationship between canopy cover and investment in reproduction in blueberries than in lingonberries [12, 34]. Similarly, because of their higher nutrient demands, blueberries may be more negatively impacted by low soil nutrients than lingonberries. Previous experimental work in the region found *V. uliginosum* showed a stronger growth response to fertilization than *V. vitis-idaea*, and in the natural system nitrogen concentrations in *V. uliginosum* are diluted

throughout the season as the plants continue to grow, but the same is not true in *V. vitis-idaea* [35,36].

Environmental variation may affect berry production indirectly through effects on pollinator abundance and activity. Pollinator and floral diversity are low in the boreal forest, many plants use multiple pollinator species, and those pollinators visit many flower species [37]. *V. uliginosum* is one of the first insect-pollinated species to flower in this habitat [pers. obs.; see also 38]. In Interior Alaska, bumblebees (*Bombus* spp.), syrphid flies (Syrphidae), and solitary bees (e.g., *Andrena* spp., and *Lasioglossum* spp.) carry the most blueberry and lingonberry pollen [39,40]. Bee genera that are present in Interior Alaska and are known to pollinate *Vaccinium* in other regions include *Osmia* spp., *Megachile* spp., and *Anthophora* [41]. High flower density around *Vaccinium* plants may lure pollinators away from the *Vaccinium* flower, as suggested by the floral market hypothesis, but could also draw pollinators into the area that otherwise would not have visited [42]. Pollen availability explained the most variation in Finnish bilberry (*V. myrtillus*) fruit production models [43] and was a limiting factor in fruit set of *V. uliginosum* in Greenland as well [44]. Between the two focal species, *V. vitis-idaea* flower structure is more adapted to cross-pollination than V. uliginosum [45]. In experiments, cross-pollination led to more fruits than self-pollination in V vitis-idaea but *V. uliginosum* had similar levels of fruit production regardless of whether cross- or self-pollinated [46–48]. Given the overall low pollinator availability in black spruce forests [37, 49], we expected plants in neighborhoods with high total floral resources to have greater pollen loads and lower pollen limitation than those in neighborhoods with low total floral resources.

Environmental conditions can also affect pollinator activity. In general, pollinators are expected to be more active in warmer sites; bees are strongly limited by temperature in Interior Alaska [50]. Bumblebees are less affected by temperature but, in Interior Alaska, solitary bees are at their lowest abundance in closed forests [51]. However, high canopy cover may also be indicative of good growing conditions for deciduous species, and result in high floral resources, resulting in a complex relationship between canopy cover and pollinator activity.

We assessed the relative effects of light (as indicated by canopy cover), depth of the active layer, soil moisture, and conspecific pollen load on flower and berry production in the boreal forest around Interior Alaska. We hypothesized that multiple variables would directly affect berry production and expected interactions among predictors. Specifically:

1. Stand history was expected to be the primary driver of environmental resource limitation: a longer time since fire was expected to result in greater light limitation.

2. Total floral resources (the number of flowers in the vicinity) was expected to have a positive influence on conspecific pollen load, and thus berry production, as a greater number of flowers in the area would attract more pollinators.

3. Blueberry ramets' relative biomass allocation to flowers and fruits was expected to be more responsive to changes in canopy cover than lingonberries' due to the differences in the two plants' life history strategies. We also expected this greater responsiveness to result in greater variability within sites.

## Methods

### Study area and site selection

About one third of the Interior Alaska boreal ecoregion is forested, with 70% of forest cover dominated by black spruce (*Picea mariana* Mill.); the rest is primarily white spruce (*P. glauca* (Moench) Voss), and deciduous trees such as Alaskan birch (*Betula neoalaskana* Sarg.),

quaking aspen (*Populus tremuloides* Michx.), and balsam poplar (*P. balsamifera* L.) [52–54]. Ericaceous shrubs such as Labrador tea (*Rhododendron groenlandicum* L.), blueberry, and lingonberry are dominant species in the understory [1]. This study focused on black spruce stands that span 13 to 200 years since last fire (stand age) and vary in slope, aspect, and forest structure to encompass a variety of growing conditions for *Vaccinium* (S1 Table). We evaluated berry production at 17 sites, each 50 m x 60 m, within the Bonanza Creek LTER Regional Site Network in the 2017 growing season (S1 Table). Previous surveys had found both blueberry and lingonberry ramets in the sites [55] and all were accessible by foot or all-terrain vehicle during early summer. No permits were necessary for the work; LTER scientists have access to the sites through agreements with the US Forest Service, State of Alaska, and the Department of Defense.

## Plant selection

Both species often reproduce clonally and grow in patches that make individual identification difficult [56]. We defined an individual, hereafter a ramet, as a single aboveground stem that did not branch within 2 cm of the soil or moss surface. From the center of each site we marked the nearest flowering *Vaccinium* ramet to a set of 12 randomly generated coordinates composed of compass degree (0–359°) and distance (0–20 m), with a search area up to 2 m. If we tagged a ramet too early in the season to distinguish fully between flower and leaf buds, and on the next visit it was clear the ramet was non-reproductive, we moved the tag to the nearest conspecific with distinguishable flower buds. Sites without flowering blueberry or lingonberry within our random points were thoroughly searched and any reproductive blueberry or lingonberry ramets were tagged. We monitored 186 blueberry ramets (mean 10.6 tagged reproductive ramets per site, range: 1–12 tagged ramets across sites) and 194 lingonberry ramets (mean: 11.3 tagged reproductive ramets per site, range: 2–12 ramets across the sites) in total. We counted flowers as they developed, and the number of berries produced when the berries at each site began to ripen. Depending on the site and the species, berry counts took place from mid-July to early August.

## Hypothesized drivers of berry production

We used five variables, measured at the site level, to investigate spatial variability and resource limitation across the landscape: elevation, active layer depth, time since fire, soil moisture, and soil temperature. Active layer depth and time since fire are both positively related to soil nutrient availability while soil moisture is negatively related to nutrient availability in Interior Alaska [57]. Much of Interior Alaska is underlain by permafrost and is water-logged, creating areas of high soil moisture and low nutrient availability due to anoxia reducing microbial activity [58,59]. N concentration in both blueberry and cranberry is greater in low-permafrost soils with a deeper active layer, and N concentrations are diluted throughout the season in blueberry (but not cranberry) as the plants continue to grow [36]. The presence of shallow permafrost cools the soil and inhibits drainage. Water collects both from weather events through the growing season as well as from the thawing ground [59]. We obtained elevation, time since fire, and active layer depth from the Bonanza Creek LTER data catalog [60]. The LTER team measured active layer depth at 20 points at each site via soil probe in the fall of 2015. We measured soil moisture (% vol; HH2, Delta-T Devices) and soil temperature (HANNA HI145) in July and August of 2018 at five points across each site, two measurements at each point (four corners and the center for a total of ten measures each visit), after 5 days without rain. Due to weather events and lack of access to sites from poor road conditions after rains, or damaged all-terrain vehicles, we could not obtain reliable soil measures in 2017. Since we were interested

in relative soil moisture and temperature between sites, we averaged the 20 measurements per site for all analyses. To measure canopy cover over each study ramet, we averaged three readings of a concave spherical densiometer measured 2 cm above the ramet, each reading taken 120˚ apart while kneeling. The above variables make up what we will refer to as "environmental variables" (elevation, active layer depth, time since fire, soil moisture, and soil temperature).

Flowers on the study plants were counted as soon as they were distinguishable, in late May to early June. Blueberry shrubs can produce over 100 flowers and do not flower all at once, so to avoid double counting on return visits we marked each flower with fabric paint. Other flowering species common in the Interior Alaskan boreal forest that overlap in flowering times include *Rhododendron groenlandicum*, *Chamaedaphne calyculata*, *Cornus canadensis*, and *Rubus chamaemorus*. We counted total floral resources, defined as all flowers of any species within 0.5 m radius of the focal blueberry or lingonberry ramet, during peak flowering of the *Vaccinium* as a measure of the potential for neighborhood plants to attract pollinators to the area or compete with focal plant flowers for resources. *R. groenlandicum* and *C. calyculata* produce many flowers per inflorescence and represent the majority of non-*Vaccinium* floral neighbors. Blueberry is one of the earliest insect-pollinated species to flower in the Interior Alaskan boreal forest [pers. obs; see also 38] The flowering timing of lingonberry overlapped with the flowering time of *R. groenlandicum* and *C. calyculata* more than with blueberry.

We estimated pollen availability by collecting two pistils from conspecific flowers near each study ramet and estimating conspecific pollen loads on the stigmas under a microscope. We attempted to collect pistils during the peak flowering period of each species. Blueberry pistils were collected between June 4 and 26, 2017 and lingonberry pistils between June 11 and 26, 2017. Pistils were mounted on microscope slides in basic fuschin gel [61] within a few days of collection. Following Spellman et al. 2015, a ramet was considered "well-pollinated" when the mean number of conspecific pollen tetrads on neighboring stigmas was >10. Blueberries produce about 45 ovules per flower and lingonberries about 32 in Interior Alaska [49] so 10 pollen tetrads (40 pollen grains) were expected to be enough for fertilization of most or all ovules. It is unknown how many ovules must be fertilized for the plant to create a fruit. We quantified fruit set as the ratio of berries to flowers on a ramet.

## Allocation measurements

Ramets, with their leaves still attached, were dried in an oven for 48 hours before leaves were removed for surface area and mass measurements. Berries from each reproductive plant were placed in a coin envelope while in the field and left in a drying oven for two weeks to ensure complete desiccation. The biomass measurements were used to assess proportion of resources allocated to leaves, stems, and berries. For each ramet we investigated reproductive and vegetative allocation by calculating the ratios of leaf mass to stem mass, flower number to leaf mass, and berry mass to leaf mass.

## Statistical analyses: Structural equation models

We expected environmental variables to be highly correlated, so to categorize the physical environment of the sites we used a principal component analysis (PCA) to sort the sites based on correlations of mean values of the environmental variables: time since fire, elevation, soil moisture, soil temperature, and active layer depth. Missing values were replaced with means from all other sites. We standardized the site averages to a mean of zero and standard deviation of one. The PCA was performed with the built-in R function princomp() [62]. Values for the first two PCA axes were used as explanatory variables in the structural equation models.

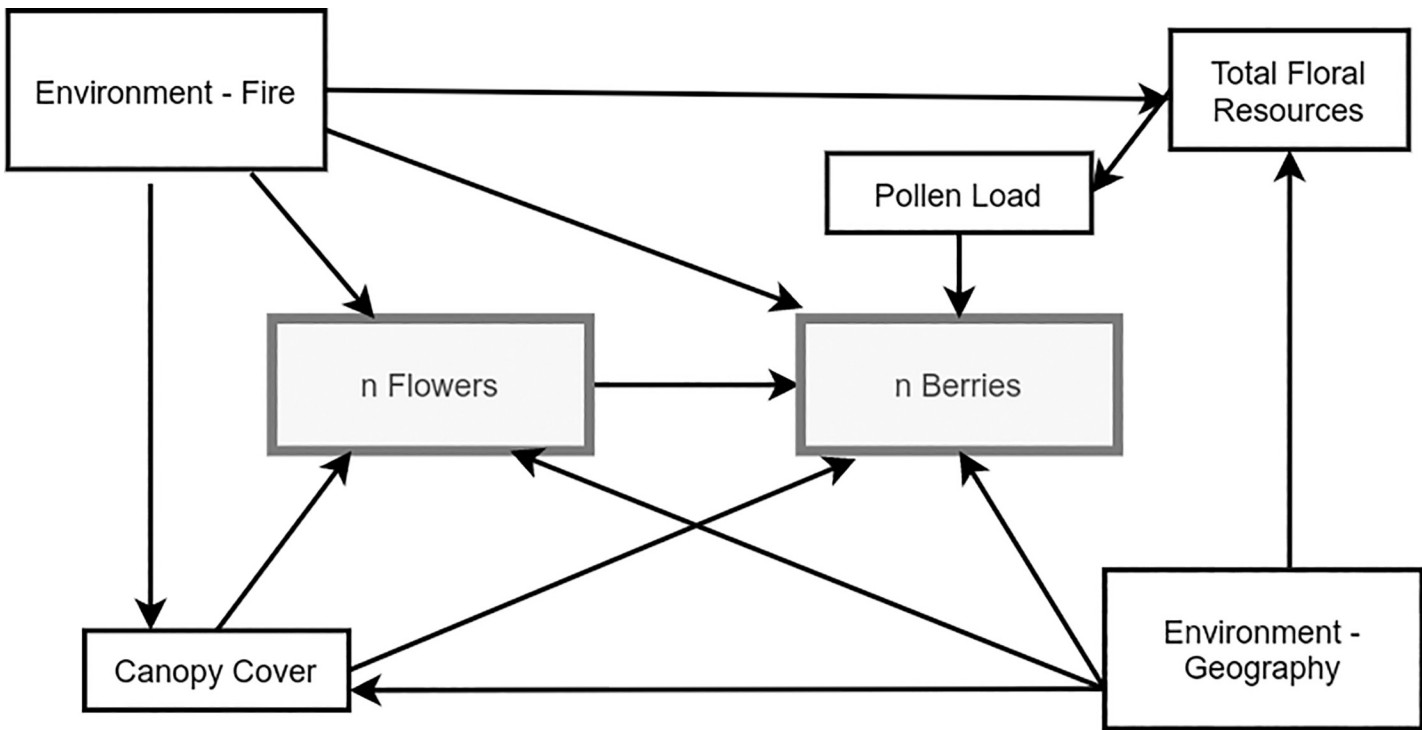

**Fig 1. Hypothesized structural equation model.** Response variables are grey. "Environment—Geography" consists of correlated factors that differ primarily by position in the landscape, "Environment—Fire" encompasses fire history. Other predictive variables were measured at each *Vaccinium* ramet.

We created a hypothetical structural equation model (SEM) [63] to assess direct and indirect effects of multiple variables on blueberry and lingonberry fruit production in 2017 (Fig 1). In our *a priori* model, we collapsed the environmental variables into two indices represented by principal components 1 and 2 (PC1 and PC2), which explained 45% and 34% of variation (Fig 2). PC1 was positively correlated with elevation and active layer depth and negatively

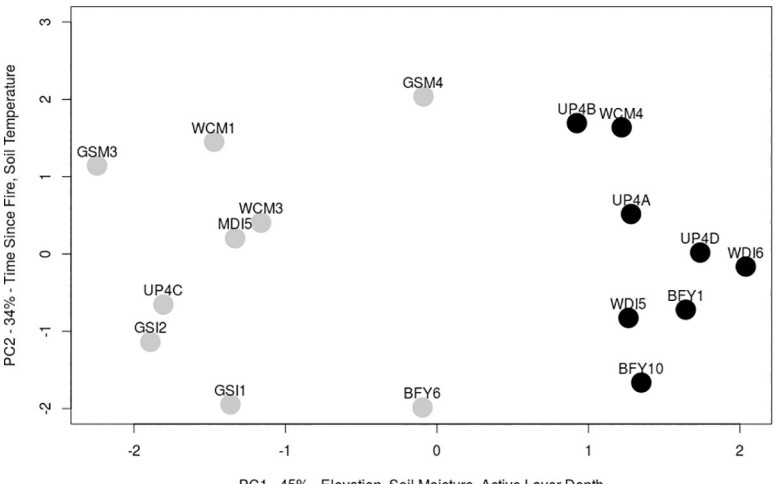

**Fig 2. PCA of environmental variables measured to encompass a 50 x 60m site.** On PC1 elevation and active layer depth were positively correlated while soil moisture was negatively correlated. Sites were divided above and below PC1 = 0 (grey and black dots) for analysis in the structural equation models, with PC1 < 0 constituting "lowland" sites and PC1 > 0 "upland" sites. Further details about the sites in S1 Table.

**Table 1. PCA loadings.**

|  | PC1 | PC2 |
|---|---|---|
| Soil temperature | - - - | **0.651** |
| Soil moisture | **-0.534** | -0.157 |
| Active layer depth | **0.563** | -0.266 |
| Time since fire | -0.186 | **0.659** |
| Elevation | **0.595** | 0.216 |

Principal components 1 and 2 were included in the structural equation model and referred to as Geography and Stand History, respectively.

correlated with soil moisture (Table 1), and reflections position on the landscape. Principle component 2 (PC2) was positively associated with time since fire and soil temperature (Table 1) and reflects site history. We used both PC1 and PC2 scores in the SEM model, calling them "geography" and" stand history" respectively.

Since number of flowers limits the number of fruits, we expected a strong positive relationship between the number of flowers produced at the beginning of the season and the number of berries. Canopy cover was expected to affect flower and fruit numbers directly through light availability but also indirectly through pollinator activity and by acting as a proxy for local growing conditions. Finally, we expected geography and stand history to influence the entire plant community in the area.

Prior to fitting the path model, we took the natural log of total floral resources, pollen load, number of flowers, and number of berries produced to improve adherence to model assumptions. For both species, we first ran a model that included data from all sites. The explanatory power for the focal response variable, number of berries produced, was low, especially for lingonberry (blueberry $R^2 = 0.31$; lingonberry $R^2 = 0.09$), suggesting that there might be differences in the direction of the relationships across site types (model in S1 Fig). To test for differences in plant responses by site type, we split our sites into two groups: one group with PC1 scores > 0 (generally higher elevation sites with low soil moisture and high active layer depth [hereafter: upland]), and the other with PC1 scores < 0 (generally low elevation with high soil moisture and low active layer depth [hereafter: lowland]) (Fig 2). Two sites (GSM4 and BFY6) were located near the center but both fell below PC1 = 0 so we grouped them with the lowland sites. Cronbach's alpha of the two PCA groups was 0.64, "Questionable" according to George and Mallery 2003 [64]. Cronbach's alpha improves to > 0.70 ("Acceptable") when we remove soil temperature or time since fire but as both are thought to be important to berry production, we sacrificed a strong group distinction in the PCA for a hopefully more explanatory SEM. We reran the models separately for each group. In all, we discuss here 6 models: an SEM for each species with ramets from all sites (2 models) and two for each species with only upland sites or only lowland sites (4 models). We also ran the models without the two intermediate sites to check the conclusions of the larger model. Without the intermediate sites, the primary influences on the response variables in the model were the same but the model fit decreased. Further manuscript analysis includes all sites (see S3 Fig for the alternate models). We did not force the regression equations in the SEM through zero to avoid interpreting beyond what we sampled. The SEMs were performed with the Analysis of Moment Structure statistical package (AMOS version 25.0) [65] which uses maximum likelihood estimation. We assessed model fit based on the ratio of minimum discrepancy to degrees of freedom (CMIN / df; ratio is between 1 and 5), root mean square error of approximation (RMSEA; lower 90% confidence interval is close to zero, 0.05 or lower), and the comparative fit index (CFI >0.90) [66].

## Statistical analyses: Allocation patterns and comparisons between species

To evaluate whether canopy cover or stand history affected allocation to reproductive vs. vegetation biomass, we ran regressions of each of the biomass ratios (leaf mass to stem mass, number of flowers to leaf mass, and berry mass to leaf mass) against canopy cover or stand history for each species (for all sites and for upland and lowland sites separately). To determine whether blueberries were more variable at the within-site level than lingonberries we calculated the coefficient of variation in flower production and berry production for each species at all 17 sites and used a Student's t-test to test for differences between the two species. All statistical analyses other than SEMs were performed using base packages in R version 3.3.2 [62].

## Results

### Pollen loads and berry production

Blueberry produced more flowers per ramet than lingonberries, and were also more variable in flower production (blueberry mean ±SD: 8.2±13.7; lingonberry: 5.1±3.3) Across all sites, 72% of blueberry flowers and 40% of lingonberry flowers were classified as well pollinated (Fig 3A; mean pollen load was 26 and 12 tetrads, respectively). For both species, upland sites had higher percentages of well-pollinated ramets than lowland sites (blueberry: 88% vs. 61%; lingonberry: 55% vs. 24%). In blueberry 24% of flowers produced fruit and in lingonberry 31% (Fig 3B). The mean number of berries produced per ramet at each site ranged from 0.08 to 9.83 (total berries per site: 0–118) and 0–1.92 (total: 0–23) for blueberries and lingonberries, respectively (Fig 3C). Upland sites produced more fruits than lowland sites for both species (Fig 3C).

### SEM model fit

Our first multi-group SEM, including all sites but divided by species, fit poorly with none of the three metrics falling in the proper range (CMIN/df of 4.290 [good fit: 1–5], CFI was 0.806

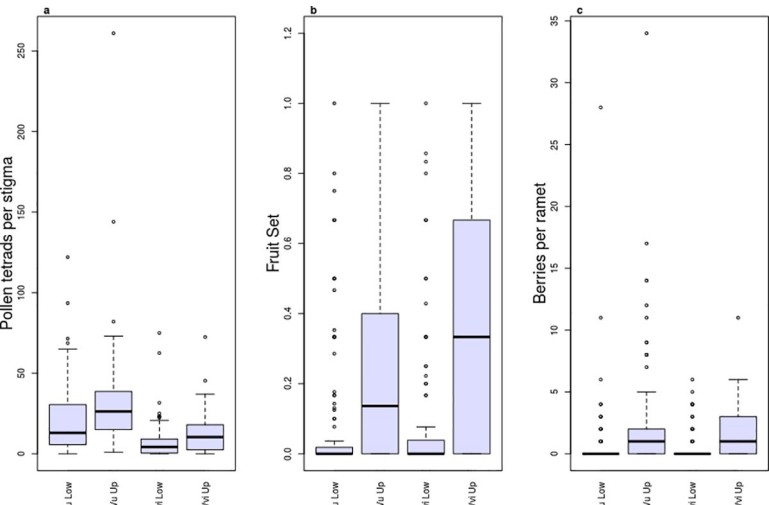

**Fig 3.** a) Pollen load, b) fruit set, and c) number of berries produced, per ramet from all blueberry (*Vaccinium uliginosum*) and lingonberry (*V. vitis-idaea*) ramets. Categories on the x-axis are *V. uliginosum* lowland, *V. uliginosum* upland, *V. vitis-idaea* lowland and *V. vitis-idaea* upland. Boxplot midline is the median, the box is the third and first quartile, the whiskers extend up to 1.5 times the interquartile range from the top of the box to the furthest datum within that distance. Grey dots are the raw data.

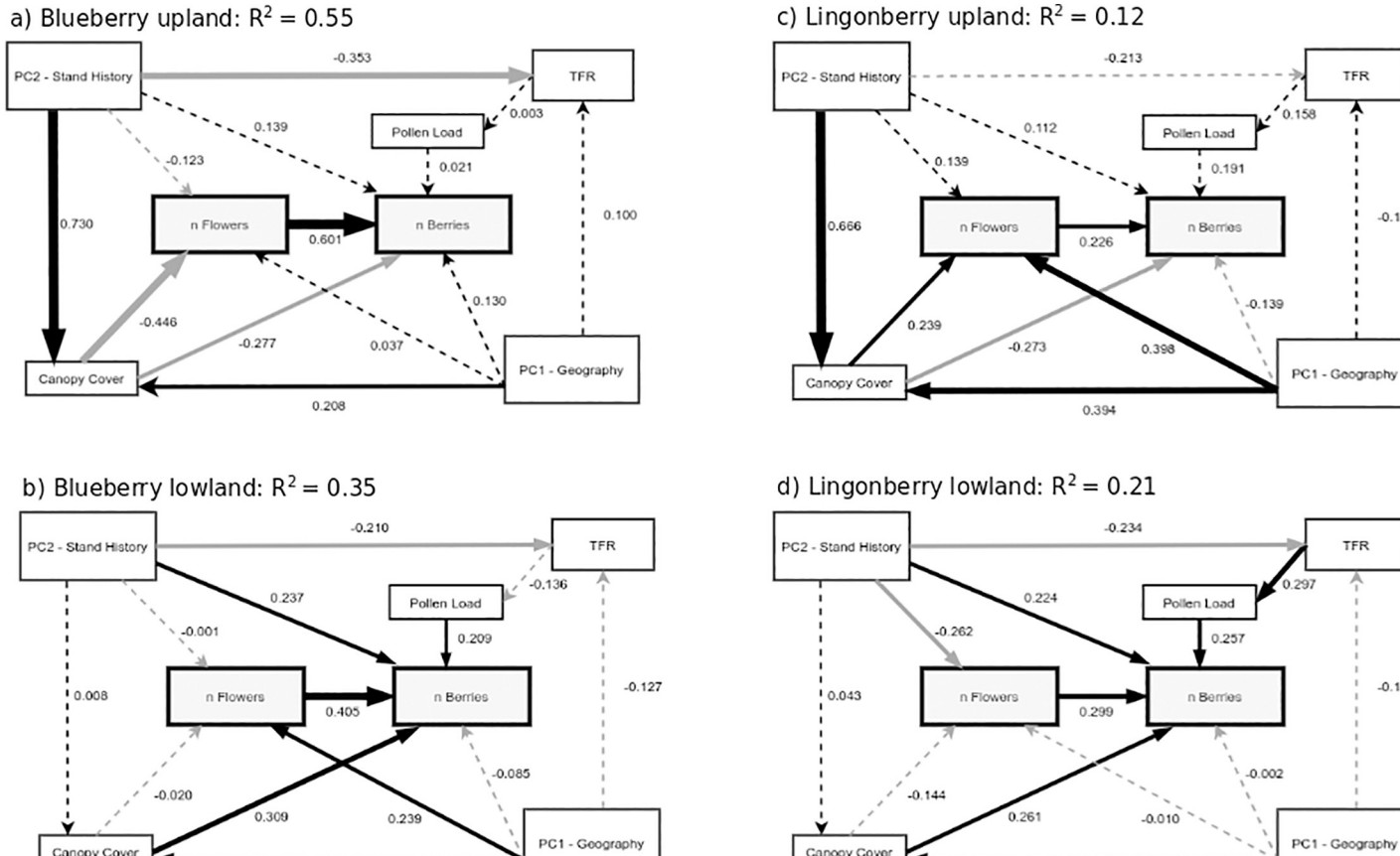

**Fig 4. Final structural equation model pathways.** a) upland blueberry, n = 80; b) lowland blueberry, n = 106; c) upland lingonberry, n = 97; d) lowland lingonberry, n = 98. Response variables outlined in bold. Solid lines represent significant pathways, while dashed lines are non-significant. Black lines represent positive pathways, while grey lines are negative pathways. Path coefficients are the standardized estimates from the SEM. $R^2$ is for the number of berries.

[fit: >0.90], and RMSEA was 0.093 [90%CI: 0.070–0.118; fit: 0.05 inclusive])(S2). The models using all sites explained 31% of the variation in blueberry fruit production but only 9% of lingonberry fruit production. When models were run after separating the data by upland and lowland sites, fit statistics improved: CMIN/df was 2.735, CFI was 0.902 and RMSEA was 0.068 (90% CI: 0.050–0.086); more paths were significant, and $R^2$ values improved (Fig 4).

## Limitations for fruit production: Flower numbers and pollen loads

As expected, flower number had a positive effect on berry number for both species, although the relationship was much stronger for blueberry than for lingonberry (S1 Fig). In contrast, pollen load had a clear positive impact on fruit number only in lingonberry (S1 Fig). However, when the sites were divided into upland and lowland, the positive relationship between berry production and pollen load was only seen in the lowland sites in both species (Fig 4). Total floral resources were only important in the lowland model for lingonberry (Fig 4D), making this the only model in which both components of the pathway from total floral resources to pollen and pollen to berries were significant. We also ran a model that included only conspecific flowers in the total floral resources measurement; the *V. uliginosum* models were the same but some relationships between explanatory variables changed in *V. vitis-idaea* changed (S2 Fig). The largest change was the influence of stand history on floral resources in both upland and

lowland *V. vitis-idaea* sites switched from negative, when all flowers were considered, to positive, when only conspecifics were included.

## Limitations for flower and fruit production: Canopy cover

In the SEM that included all sites the only significant effect of canopy cover was on blueberry flower production (a negative relationship; S1 Fig). Upland sites in our study had higher canopy cover than lowland sites (45 ± 20% vs. 31±17% for blueberry sites and 49±19% vs. 35±18% for lingonberry sites; $F_{(1,350)}$ = 51.6, P < 0.001 for all sites combined). When upland and lowland sites were evaluated separately, canopy cover had differing effects on flower production depending on the species (Fig 4). In upland sites the relationship between canopy cover and flower number was negative in blueberries and positive in lingonberries while in lowland sites there was no relationship for either species. Canopy cover negatively influenced berry number at upland sites and positively influenced berry number in lowland sites for both blueberry and lingonberry (Fig 4). We reran models without the two intermediate sites, BFY6 and GSM4, to verify results, however, this reduced model fit. Even without the data from the two middle sites the relationship between canopy cover, flowers, and berries in lowland conditions remained largely the same (see full results in S3 Fig).

## Direct and indirect effects of stand history and geography

The only direct effects of stand history (PC2) were a positive relationship with flower production for the lowland lingonberry ramets and a positive relationship with berry production in lowland sites for both species (Fig 4, S1 Fig), indicating that plants in older (burned longer ago) lowland sites were more productive. However, stand history had indirect effects as well: stand history was strongly positively related to canopy cover in upland sites, and negatively related to total floral resources in five out of six models (all except upland sites for lingonberry; Fig 4). As a result, the total impact of stand history was positive for lowland sites in both species (where positive direct effects outweighed negative indirect effects), but negative for upland blueberry sites and neutral for upland lingonberry sites (where negative indirect effects outweighed or balanced positive direct ones; Table 2). Geography (PC1) had no clear direct impacts on berry production, but indirect positive effects via flower production in lowland blueberry and upland lingonberry sites, and additional indirect effects via positive

**Table 2. Direct and indirect effects on number of berries on SEMs by species and landscape type shown in Fig 4 in order of the absolute value of the total effect.**

|  | Blueberry | Direct | Indirect | Total | Lingonberry | Direct | Indirect | Total |
|---|---|---|---|---|---|---|---|---|
| **Upland** | Flowers | 0.601 | -- | 0.601 | Flowers | 0.226 | -- | 0.226 |
|  | Canopy | -0.277 | -0.268 | -0.545 | Canopy | -0.273 | 0.054 | -0.219 |
|  | Stand History | 0.139 | -0.324 | -0.184 | Pollen | 0.191 | -- | 0.191 |
|  | Geography | 0.130 | -0.091 | 0.039 | Geography | -0.139 | 0.004 | -0.135 |
|  | Pollen | 0.021 | -- | 0.021 | TFR | -- | -0.030 | -0.030 |
|  | TFR | -- | < 0.001 | < 0.001 | Stand history | 0.112 | -0.121 | -0.009 |
| **Lowland** |  |  |  |  |  |  |  |  |
|  | Flowers | 0.405 | —0.405 | 0.405 | Flowers | 0.299 | -- | 0.299 |
|  | Canopy | 0.309 | -0.008 | 0.301 | Pollen | 0.257 | -- | 0.257 |
|  | Stand History | 0.237 | 0.008 | 0.245 | Canopy | 0.261 | -0.043 | 0.218 |
|  | Pollen | 0.209 | -- | 0.209 | Stand history | 0.224 | -0.051 | 0.173 |
|  | Geography | -0.085 | 0.189 | 0.103283 | Geography | -0.002 | 0.066 | 0.065 |

"Geography" refers to PC2 scores and "Stand history" to PC1 scores. "TFR" is total floral resources. Dashes indicate this link was not assessed in the model.

**Table 3. The relationships between biomass ratios and canopy cover for all blueberry (*Vaccinium uliginosum*) and lingonberry (*V. vitis-idaea*) ramets.**

| Response variable ratios | Blueberry | Lingonberry |
|---|---|---|
| Leaf mass: stem mass | -0.0002, $p = 0.71$ | **-0.0213, $p < 0.001$, $R^2 = 0.11$** |
| n Flowers: leaf mass | -0.0893, $p = 0.084$ | **-0.4362, $p = 0.003$, $R^2 = 0.04$** |
| Berry mass: leaf mass | -0.0005, $p = 0.20$ | 0.0004, $p = 0.49$ |

Parameter estimate (correlation coefficient) and $p$ values for all relationships. Significant ($p < 0.05$) relationships are in **bold** and contain the adjusted $R^2$ value. Blueberry n = 186; lingonberry n = 205.

relationships with canopy cover in upland sites (Fig 4). Opposing effects resulted in weak total relationships between geography and berry production for all four models (Table 2).

## Most important drivers

When looking at all sites, flower production had the greatest impact on blueberry production and pollen load on lingonberry production (S2 Table). When ramets were split into upland and lowland sites, flower production was the most influential variable in all models, but other drivers differed (Table 3). For blueberry, canopy and stand history were second and third, but with opposite directions for upland sites (negative) and lowland sites (positive). For lingonberry, canopy cover and pollen loads were second or third, again with opposite directions for canopy cover (negative in upland sites, positive in lowland sites).

## Biomass allocation given canopy cover

When ramets from all sites were included, relationships between canopy cover and allocation patterns were weak (Table 3), with only allocation to leaves (as measured by leaf mass to stem mass) showing an $R^2 > 0.10$ (sites with higher canopy cover have lower allocation to leaves). In contrast, when we divided the ramets into the upland and lowland groups, there were multiple relationships for upland sites (Table 4A). In both blueberries and lingonberries in investment in leaves relative to stems decreased as canopy cover increased, while investment in berries relative to leaves decreased (Table 4A). Plants in lowland sites showed little change in allocation with canopy cover (Table 4B).

**Table 4. The relationships between biomass ratios and canopy cover for blueberry (*Vaccinium uliginosum*) and lingonberry (*V. vitis-idaea*) ramets in upland and lowland sites.**

| a) Upland | | |
|---|---|---|
| Response variable ratios | Blueberry | Lingonberry |
| Leaf mass: stem mass | **-0.0015, $p = 0.003$, $R^2 = 0.10$** | **-0.0079, $p < 0.001$, $R^2 = 0.24$** |
| n Flowers: leaf mass | -0.0584, $p = 0.342$ | 0.0367, $p = 0.73$ |
| Berry mass: leaf mass | **-0.0021, $p < 0.001$, $R^2 = 0.16$** | -0.0014, $p = 0.205$ |
| b) Lowland | | |
| Response variable ratios | Blueberry | Lingonberry |
| Leaf mass: stem mass | 0.0009, $p = 0.364$ | -0.006, $p = 0.444$ |
| n Flowers: leaf mass | -0.0649, $p = 0.476$ | **-0.6259, $p = 0.0391$, $R^2 = 0.04$** |
| Berry mass: leaf mass | 0.0006, $p = 0.348$ | 0.0007, $p = 0.280$ |

Parameter estimate (correlation coefficient) and $p$ values for all relationships. Significant ($p < 0.05$) relationships are **bold** and contain the adjusted $R^2$ value. Upland blueberry, n = 80; lowland blueberry, n = 106; upland lingonberry, n = 97; lowland lingonberry, n = 98.

### Differences between species

Canopy cover explained substantial variation of the change in allocation to leaves in lingonberry but not in blueberry ($R^2 = 0.24$ vs. $R^2 = 0.10$). However, flowering rates in blueberry decreased rapidly with canopy cover (Fig 4A) while lingonberry flowering rates were unresponsive to canopy cover (Fig 4B).

The variation in production of flowers and berries differed considerably across all 17 sites (S2 Table). However, while variation in flower production was greater in blueberries than in lingonberries (coefficient of variation: 0.87 vs. 0.33; $t = -5.79$, $p = < 0.001$), there was less evidence for a difference in variability in berry production (coefficient of variation: 1.57 vs. 0.79; $t = -1.82$, $p = 0.096$).

## Discussion

Our primary goal was to assess pollen versus resource (light and nutrient) limitation on berry production of blueberry and lingonberry across the landscape in black spruce of Interior Alaska. We found that the most important drivers of berry production differed between habitats and species. In general, lower elevation, wetter sites with shallower active layers tended to be more pollen limited than the upland, drier sites, while canopy cover was a strong predictor of berry production and allocation in upland but not lowland sites. Also, lingonberry plants tended to be more pollen limited than blueberry plants. These results suggest that the expected changes in boreal forest fire regime and subsequent effects on vegetation composition and soils are likely to have different impacts on productivity of blueberry and lingonberry, and different impacts in upland versus lowland habitat.

### Pollen limitation

Lingonberries were more pollen limited than blueberries, especially in lowland sites (Table 3). Lingonberry is partially self-incompatible [48], may be more dependent on pollinators for fertilization and suffer from geitonogamous pollination more than blueberries. In experiments, *V. uliginosum* produces the same number of fruits whether the experimenters self-pollinated or cross-pollinated the plants. However, *V. uliginosum* will not self-pollinate in the absence of visiting pollinators [45]. Factors other than self-incompatibility may also play a role: a much higher proportion of blueberries than of lingonberries were "well-pollinated" (pollen loads large enough to potentially fertilize all ovules), suggesting that either blueberries are more attractive to pollinators than lingonberries, or that they are more likely to occur in areas with high pollinator abundance. Additionally, flower number was more important in blueberry than in lingonberry in driving berry number, but that is likely simply the result of the much greater variability in flower number.

Both species showed stronger evidence for pollen limitation in lowland sites than upland, and for lingonberry the total floral resources (number of flowers of all species in the immediate area) also played a positive role in lowland sites. Previous work by Spellman et al. [49] found canopy cover, total floral resources, and air temperature were all important in explaining *V. vitis-idaea* pollination rates in black spruce sites but not mixed deciduous sites (analogous to our lowland and upland delineations). Overall, the results reinforce the idea that in cold, wet habitat the low abundance of pollinators limits fruit production.

### Resource limitation

In upland sites canopy cover was a strong negative influence on both blueberry and lingonberry fruit production. High canopy cover may result in light limitation, though competition

from tree and shrub species for nutrients and water may also play a role. The increased investment by both species in stems relative to leaves in upland sites as canopy cover increased is consistent with greater competition for light. The positive correlation between canopy and berry production in lowland sites for both species may be the result of relatively less investment in leaves, leading to higher productivity in both *Vaccinium* and its neighboring species. This is supported by the higher blueberry fruit set (the ratio of berries to flowers) with higher canopy cover at low elevation sites and hierarchical regressions in which the biomass investments of lingonberry ramets were significantly related to mean canopy cover of the site level but not the canopy cover immediately above the ramet. Overall, these results suggest that light limitation plays a larger role in upland sites, while nutrient limitation (the result of cold, wet soils) plays a larger role in lowland sites.

In upland blueberries, high canopy cover not only reduced berry number but also flower number. This is consistent with flower production of globe huckleberry (*V. globulare*) in Montana, where reduced flower numbers were attributed to resource limitation above 30% canopy closure [67]. Surprisingly, the relationship between canopy cover and flowers in upland lingonberries was positive, thereby somewhat mitigating the negative direct effects on berry number. However, the negative relationship between canopy cover and blueberry flowers was almost twice the strength of the positive relationship between upland canopy cover and lingonberry flowers. Though we didn't directly measure pollinators, many other studies have also found higher canopy cover leads to lower abundance and activity levels of pollinators, leading to an indirect effect on pollen limitation [51]. Stand history had a significant, positive effect on canopy cover in upland sites—the longer it had been since a fire, the more shrubs and trees had grown around the *Vaccinium*—but no relationship with canopy cover in lowland sites. This suggests that a major driver of variation in canopy cover in upland sites is successional stage, while in lowland habitat other factors (such as local drainage conditions) drive variation. The different regeneration patterns were also found in the boreal black spruce forests of Quebec [68]. In that study, less productive sites, those with excess moisture, had slower rates of regeneration and a different community structure post fire than sites with drier conditions. Increased fire frequency may have a positive effect on blueberry productivity in upland sites (through a reduction in canopy cover) but not in lowland sites (where older and high canopy-cover sites were most productive) nor in lingonberry (where stand history had a minimal impact).

### Differences between species in responsiveness

We had predicted that blueberries would be more responsive to variation in the environment than lingonberries because of their more resource-acquisitive life history and shorter leaf lifespan. This was supported by the greater variability in flower number: blueberries were more limited by flowers production than lingonberries and had a much stronger relationship between the proportion of ramets that were reproductive (produced at least one flower) and canopy cover (Fig 5). Blueberry responsiveness was also reflected in the greater ability of the SEMs to explain variation in berry production in blueberries. However, the importance of pollen limitation for lingonberry, especially in lowland sites, likely accounts for the much smaller difference between the two species in variability in berry production.

### Study limitations

All measurements, except soil moisture and temperature, took place in a single growing season (in 2017). The 2017 growing season in Interior Alaska was warmer than normal, with temperatures in 2017 well above the long-term average for June (17.1˚C vs. mean 15.4 ± 1.5 (SD) ˚C

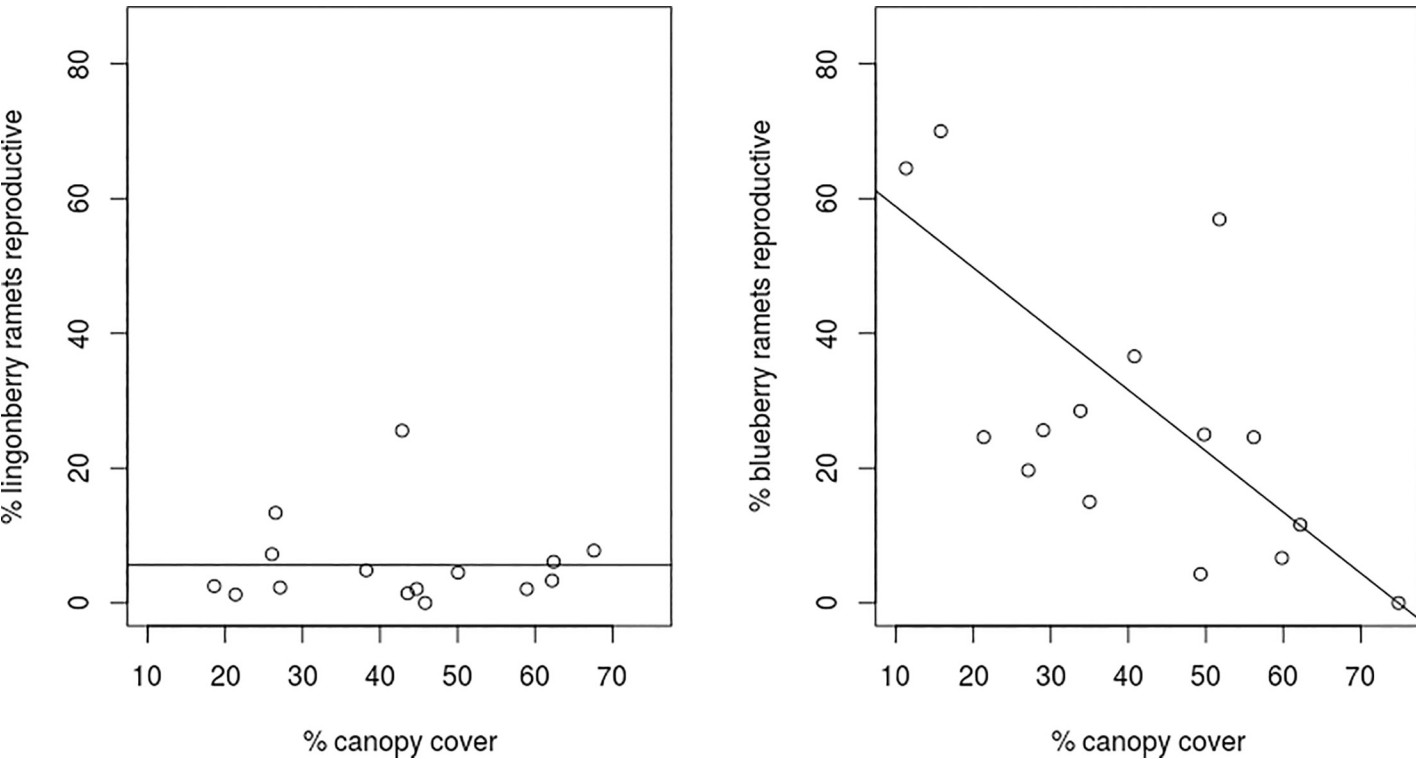

**Fig 5. The percentage of reproductive ramets at each site by the percentage of canopy cover at the site.** a) blueberry (*V. uliginosum*) $p = 0.007$, $R^2 = 0.37$, n = 15; b) lingonberry (*V. vitis-idaea*) $p = 0.998$, n = 15. Sample size was reduced because no flowering plants were found at one site for each species.

for 1941–2010) and July (18.7˚C vs. 16.6 ± 1.3˚C) (data from the Fairbanks International Airport obtained via the National Centers for Environmental Information). Furthermore, June was wetter than usual by about 25% (43.9 mm in total, compared to a long-term average of 35.1 mm). Interannual variation in temperature and precipitation are likely to affect pollinator activity and resource limitation. However, while this may result in changes in the absolute roles of these variables (e.g., a site that was not limited by pollinators in 2017 may be limited in a colder or wetter year), we expect the relative importance in the two different habitats (greater resource limitation in upland sites and greater pollinator limitation in lowland sites) and for the two different species (greater pollen limitation in lingonberry) to be consistent across years.

Both blueberry and lingonberry are clonal, but lingonberry can form dense mats of genetically identical ramets that share resources [1]. The trade-offs between vegetative and reproductive growth may not be occurring within a single ramet but across many connected ramets in an area. Additionally, lingonberry leaves last for several years, so trade-offs in allocation to leaves vs. flowers or fruits under changing environmental conditions are likely to be difficult to detect when all leaves are included in the analysis (as in this study). Future work should focus on trade-offs between flower and leaf initiation, both of which take place a year before flowering and leaf-out (CPH Mulder, pers. obs.) or between fruit production and leaf production in the following year.

Patterns in berry production and resource allocation were stronger in the upland sites. Lowland sites encompassed a greater range of site conditions, so it appears that environmental limitations were driven by something we missed in our study. Studies of *Vaccinium* species and boreal plant communities in Sweden have found soil pH and soil microbial activity play a

role in community composition and *Vaccinium* biomass allocation [69,70]. Additionally, Interior Alaska contains a variety of wetland types with different combinations of water movement, soil type, and permafrost, all of which affect the plant communities above them [71,72]. Future work in Interior Alaska to elucidate the controls on *Vaccinium* productivity should follow examples of studies in Fennoscandia by including soil pH, direct measures of nutrient cycling, and wetland conditions.

## Potential changes in berry production under an altered fire regime

The significance of canopy cover on berry production in the uplands leads to two potentially contrasting outcomes for future berry production in Interior Alaska. The change in forest fire dynamics caused by climate change is leading to an increase in fire size, severity, and frequency [14, 27,28]. The increase in size and frequency will lead to a lower median stand age, generating canopy cover in the range most conducive to berry production ($<$ 30%). Research in Russia and Montana has found berry production peaks 10–20 or 25–60 years after a wildfire, respectively [67, 73]. Upland sites could see an increase in berry production under lower canopy cover. However, lowland sites may still be limited by pollinator abundance or other resources not associated with canopy. Additionally, fires are also changing in severity. More severe fires consume not just the plant communities above the soil but much of the soil organic layer itself [27], changing the immediate and long-term successional dynamics of the forest [74]. More severe and more frequent fires create a new stable state of succession that instead of transitioning from hardwoods to spruce stays hardwood until the next fire [75–77].

In summary, our results show that both resource limitation and pollen limitation play a role in limiting fruit production of blueberries and lingonberries, with light limitation being a primary factor in upland sites while pollen limitation is important in lowland sites. We recommend that models predicting productivity under a changing climate incorporate pollinator availability as well as changes in resources.

## Supporting information

**S1 Fig. Structural equation model pathways for all sites combined.** a) All blueberry (*Vaccinium uliginosum*) ramets, n = 186; b) All lingonberry (*V. vitis-idaea*) ramets, n = 195. Response variables outlined in bold. Solid lines represent significant pathways (p$<$0.05), while dashed lines are non-significant. Black lines represent positive pathways, while grey lines are negative pathways. Path coefficients are the standardized estimates from the multi-group structural equation model. $R^2$ is for the number of berries.
(TIFF)

**S2 Fig. SEM pathways when total floral resources is replaced with conspecific floral resources.** a) high elevation blueberry (*Vaccinium uliginosum*), n = 80 b) low elevation blueberry, n = 106 c) high elevation lingonberry (*V. vitis-idaea*), n = 97 d) low elevation lingonberry, n = 98. Grey boxes are the response variables. Solid lines represent significant pathways, while dashed lines are non-significant. Black lines represent positive pathways, while grey lines are negative pathways. Path coefficients are the standardized estimates from the SEM.
(PDF)

**S3 Fig. SEM pathways when sites BFY6 and GSM4 are removed from the lowland designation.** Fit statistics worsened a) high elevation blueberry (*Vaccinium uliginosum*), n = 80 b) low elevation blueberry, n = 106 c) high elevation lingonberry (*V. vitis-idaea*), n = 97 d) low elevation lingonberry, n = 98. Grey boxes are the response variables. Solid lines represent significant pathways, while dashed lines are non-significant. Black lines represent positive pathways,

while grey lines are negative pathways. Path coefficients are the standardized estimates from the SEM.
(PDF)

**S1 Table. Site names and descriptions.** Upland sites are highlighted grey, lowland sites are white.
(DOCX)

**S2 Table. Direct and indirect effects on number of berries in each SEM shown in S2 in order of the absolute value of the total effect.** "Geography" refers to PC2 scores and "Stand History" to PC1 scores. "TFR" is total floral resources. Dashes indicate this link was not assessed in the model.
(PDF)

**S3 Table. Coefficient of variation in flowers and berries produced for blueberries (*Vaccinium uliginosum*) and lingonberries (*V. vitis-idaea*) in each site.**
(DOCX)

**S1 Raw data.**
(CSV)

## Acknowledgments

We thank Jamie Hollingsworth, Mark Winterstein, Laila Brubaker, Tessa Hasbrouck, Alice Ramos de Moraes, Sean Lyons, and Eleanor Lynch for their help in the field, and Katie Spellman, Teresa Hollingsworth, and Diane Wagner for their constructive comments on earlier drafts.

## Author Contributions

**Conceptualization:** Lindsey Viann Parkinson.

**Formal analysis:** Lindsey Viann Parkinson.

**Funding acquisition:** Christa P. H. Mulder.

**Investigation:** Lindsey Viann Parkinson.

**Methodology:** Christa P. H. Mulder.

**Project administration:** Christa P. H. Mulder.

**Resources:** Christa P. H. Mulder.

**Supervision:** Christa P. H. Mulder.

**Writing – original draft:** Lindsey Viann Parkinson, Christa P. H. Mulder.

**Writing – review & editing:** Lindsey Viann Parkinson, Christa P. H. Mulder.

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
