## [Decision Letter · Decision Letter 0]

23 Dec 2019

PONE-D-19-27691

Spatial variability in pollen and resource limitation for fruit production of two species in Interior Alaska: Vaccinium uliginosum and V. vitis-idaea

PLOS ONE

Dear Dr Parkinson,

Thank you for submitting your manuscript to PLOS ONE. After careful consideration, we feel that it has merit but does not fully meet PLOS ONE’s publication criteria as it currently stands. Therefore, we invite you to submit a revised version of the manuscript that addresses the points raised during the review process.

In general, the two reviewers considered your manuscript a useful and scientifically sound contribution to the topic, however there are some issues that have to be addressed in a revised version.

Both reviewers found that some important references were missing. This refers to the assumptions and hypotheses underlying your study, but also to the species’ reproductive biology and to pollen limitation at high latitudes in general. Some other issues that require more attention are the role that vegetation cover and light availability may indirectly play for seed set via their effects on pollinator activity and the role of pollen “quality” limitation caused by self-incompatibility and extensive clonal spread.

The reviewers also raised a range of questions regarding the methods used. Among others, I recommend clarifying what you considered a ramet and why you used a different radius for the two species to count floral resources, adding some more information about soil nutrient availability of sites (e.g. leaf nutrient concentrations) if possible, and improving the assignment of sites to the groups, in particular of the two rather intermediate sites. The reviewers also recommended some adjustments of statistical methods, e.g. using multilevel modeling because of the nested data structure, and adding some missing but important relationships to the SEMs.

Furthermore, I recommend considering the further, detailed comments provided by the two reviewers, including suggestions how to improve writing, formatting and figures.

We would appreciate receiving your revised manuscript by Feb 04 2020 11:59PM. To enhance the reproducibility of your results, we recommend that if applicable you deposit your laboratory protocols in protocols.io, where a protocol can be assigned its own identifier (DOI) such that it can be cited independently in the future. For instructions see: http://journals.plos.org/plosone/s/submission-guidelines#loc-laboratory-protocols

We look forward to receiving your revised manuscript.

Kind regards,

Harald Auge

Academic Editor

PLOS ONE

Journal Requirements:

2. In your Methods section, please provide additional location information of the sampling sites, including geographic coordinates for the data set if available.

3. In your Methods section, please provide additional information regarding the permits you obtained for the work. Please ensure you have included the full name of the authority that approved the sampling sites access and, if no permits were required, a brief statement explaining why.

4. We note that Figure 1 in your submission contains map images which may be copyrighted.

Reviewers' comments:

Reviewer's Responses to Questions

**Comments to the Author**

1. Is the manuscript technically sound, and do the data support the conclusions?

Reviewer #1: Yes

Reviewer #2: Yes

2. Has the statistical analysis been performed appropriately and rigorously? 

Reviewer #1: Yes

Reviewer #2: Yes

3. Have the authors made all data underlying the findings in their manuscript fully available?

Reviewer #1: Yes

Reviewer #2: Yes

4. Is the manuscript presented in an intelligible fashion and written in standard English?

Reviewer #1: Yes

Reviewer #2: Yes

5. Review Comments to the Author

Reviewer #1: In the study, the relative effects of light (canopy cover), nutrient resources (depth of the active layer and soil moisture), and pollen availability (pollen load) were assessed on flower and berry production in the boreal forest around Interior Alaska. The study is a useful contribution, and a great effort was given to study a complex system of factors affecting on berry plants. The use of structural equation model (SEM) is well justified to analyse relationships among unobserved concepts using the above mentioned variables observed. Data collected in a single year only is understandable (due to high work load in the field and lab) and is not limiting the value of the study. I have comments, some of which may be regarded as pointing some shortcomings.

1. In Introduction, the authors present several assumptions without any references (“we expect”, “may “be, “may affect”, etc.). I suggest that you try to find references to support your expectations/assumptions, and separate your hypotheses from the results published in literature. For example, Finnish studies by Anne Tolvanen and Jussi Kuusipalo (1988, Vegetation 76: canopy cover and fruiting of bilberry) may support some of your speculations and results.

2. According to the caption of Fig. 3, the size of the site seems to be 50 m x 60 m. This should be mention already in Methods.

3. In Methods (and Abstract), please mention as early as possible that the measurements took place in a single growing season in 2017.

4. You assessed flowers of other species flowering at the same time as Vaccinium species studied. What about species flowering earlier in the spring, and thus luring and sustaining pollinators at the site. Should you assess these species, especially willows as well?

5. Any reason to count total floral resources using different radius for blueberry (1 m) and lingonberry (0.5 m)?

6. In PCA, you could calculate and present Cronbach’s alpha if variable deleted to explore the internal consistency of the variables included in PC analysis.

7. Please use consistently “Age” or “Time since fire”. In Table 1, you use Age, and in Fig. 3, Time since fire.

8. “Canopy cover explained more than twice as much of the change in allocation to leaves in lingonberry as it did in blueberry (R2 =0.24 vs. R2 =0.10).” I am not sure if you can compare two R2 values obtained for two different data sets.

9. I guess simple linear regression models (no equations were presented) were fitted to explain the biomass ratios as a function of canopy cover. You have a hierarchical data structure; ramets within sites? Therefore, you should use a multi-level modelling in fitting the regression equations.

10. The number of observations is missing in Tables 4 and 5. In Fig 7, the number of observations is 15. Why? The number of sites was 17.

11. The authors discuss on possible impacts of the expected changes in boreal forest fire regime, vegetation composition and soils on berry productivity. I am not aware of forest management in Alaska. If managed, then the effect of forest management (affecting the canopy cover) on berry production should also be discussed.

Reviewer #2: Journal: PLoS ONE

Manuscript ID: PONE-D-19-27691

Title: Spatial variability in pollen and resource limitation for fruit production of two species in Interior Alaska: Vaccinium uliginosum and V. vitis-idaea

Comments to the Editor and Authors

General:

This study explores how site variation relates to flower and fruit production in two species of Vaccinium in Alaska. Specifically the study tests predictions of patterns in abiotic resource and pollen limitation in fruit production related to time since last fire, total local floral resources, and differences in the two species’ life histories. The primary results are analyzed using structural equation modeling that offer insights into direct and indirect effects on flower and fruit production and differences between upland and lowland habitats. Overall the results are in accordance with expectations, although the authors found evidence for total floral resources affecting pollen loads or the number of berries in only one of four contexts.

In general, the underlying questions are of ecological interest, the methods and the analyses appear sound to my knowledge (though I have one or two questions), and the interpretation of results is appropriate. The grounding of research in fruit limitation research could be strengthened. The writing, formatting, and figures could be substantially improved. Overall, I believe it is a fine study and sufficient manuscript to warrant publication with some improvements and questions addressed. I provide some general statements and questions followed by specific edits/suggestions below.

I recommend the authors develop a stronger foundation of the study species’ reproductive biology and pollen limitation on fruit production in high latitude contexts. Interpretation of their results would be improved information by providing context to highly relevant studies that are not referenced – e.g., Jacquemart & Thompson 1996; Jacquemart 1997; Brown & McNeil 2009, a number of Totland’s papers; Fulkerson et al. 2012; Urbanowicz et al. 2018. For example, V. uliginosum is self-compatible where it has been studied, but does not auto-pollinate in the absence of visitors. Often open pollination treatments result in equivalent proportions of fruit set (but not seed set) as supplemental hand pollinations. Pollen limitation and resource limitation of reproduction are typically measured as a proportion of fruits to flowers or seeds to flowers (or ovules) under experimentally enhanced and control conditions - there is a value in the sense that the authors use the terms and by using the natural environmental variation, but referencing the distinction would be useful.

I would recommend that the authors acknowledge pollen limitation may be a function of both pollen quantity and pollen “quality” (i.e., amount of outcrossed pollen). Pollen quality limitation can be an issue especially for highly clonal plants even with relatively abundant pollinators. Vaccinium vitis-idaea is self-incompatible and therefore more likely to be subject to geitonogamous pollinations that don’t result in fruits or seeds.

While I concur with the general relationship between active layer depth, soil moisture, and nutrient availability, however that relationship is noisy and can be variable within sites. Thus we are presented with a proxy for soil nutrient availability with unclear relationship to what the plants actually have access to. The authors recognize this and briefly discuss it in the Methods, which is appropriate. If tissue from the ramets has been retained following the biomass measurements, could leaf N content or concentration at least be measured at least? Has anyone else measured soil N and P at these LTER sites? Obviously it is too late in the year to get soil samples now.

Some more description of what the authors consider a ramet for both of these species would be warranted in the Methods, as what constitutes the “ground” in boreal forests with deep layers of loose organic layers or moss can be ambiguous, especially coupled with the extensive branching of these species low to the ground.

It should be stated in the pollen availability section that the authors were only counting conspecific tetras or no heterospecific tetrads were observed.

The authors should note throughout the manuscript that canopy cover/light availability not only may affect flower and fruit production through physiological factors, but also due to increased pollinator activity – there should be plenty of references of that in the literature.

It would make more sense in my mind to drop the two sites (GSM4 and BFY6) that were intermediate in environmental characteristics as described by the PCA rather than assigning them a lowland category. Or I suppose a hierarchical cluster analysis or another analysis that assigns groups could be a more defensible approach for determining where the two sites should go.

The SEM has some relationships that seem to me to be lacking and potentially important. Why would there not be estimation of the effect of # Flowers on Pollen Load? Only Total Floral Resources on Pollen Load? It is likely of much less importance how many Rhododendron and other flowers are present on pollen loads than number of Vaccinium flowers. What about the relationship between canopy cover and total floral resources?

There are a lot of SEM figures and the first model could be included as a figure only in the supplemental figures section. Removing it from the primary manuscript would improve the clarity of the message.

There are some sections of the manuscript that read smoothly and the overall architecture of paragraph succession is logical, but much of it is packed with unnecessary parenthetical clauses, missing punctuation, sentences that could be improved for clarity, and poor overall formatting. I would encourage the authors to spend a bit more time and carefully craft the language and formatting. I will include a few suggestions below in the specific section, but the number of edits would be too lengthy to include all corrections or suggestions.

I appreciate the opportunity to read this paper and I sincerely hope that my comments are useful in generating a more robust manuscript. As with all papers I am asked to review, I strive to provide honest, thorough, and fair reviews, I therefore do not have cause to maintain anonymity.

-Matthew L. Carlson

Specific:

Figure 1: I would remove this figure. You include a supplemental table that could include latitude longitudes as well. If you do feel it is necessary it could use some cartographic/design attention (both the inset and the primary maps are very difficult to discern features and the figure legend does not do an adequate job of explaining what the reader is looking at.

Figure 4: Please remove the raw data points. The inclusion of the points makes the figures unnecessarily busy and distracting. It is not clear from the axis label or legend what “Pollen Load” is (pollen grains/stigma or pollen tetrads/stigma… not pollen grains/ramet as suggested by the legend?). Also I would suggest the y-axis label of # Berries, say Berries per ramet.

Title: “Spatial variability…” is misleading since there is a whole subdiscipline of ecology that explores spatial patterns and variability and this study does not explicitly address space. Something along the lines of: “Patterns of pollen and resource limitation of fruit production in Vaccinium uliginosum and V. vitis-idaea in interior Alaska” or try and squeeze in some primary variables: “Pollen and resource limitation of fruit production in Vaccinium uliginosum and V. vitis-idaea in interior Alaska: effects of landscape position, stand age, and canopy cover on reproduction”

Abstract: Looks okay other than far too many parentheses and in the last sentence it is really pollination service or pollinator service, not the raw abundance of pollinators that is important in fruit production.

Introduction:

Paragraph 1 -

Missing a period in the first sentence.

Avoid the long list of common names – the single more often used one would be appropriate.

“Ursus spp.” should be “Ursus spp.” or just say Ursus arctos and U. americanus. There are only two after all.

Branta hutchinsii reference to Hupp’s paper – those are a more coastal – nesting species that I believe he was emphasizing feed on Empetrum. Maybe I’m wrong and he did talk about them feeding on Vaccinium.

Paragraph 2 –

“…pollinator availability” I would recommend changing to “pollinator service (limiting pollen compatible deposition and subsequent fertilization)”

I would strike (i.e. where on the landscape we would expect changes in berry production)

“…growth and fruit availability are not always correlated” I would say “… growth, fruit, and seed production are not always correlated with one another [21].”

“Fruit production in all plants…” should read “Fruit production in all flowering plants…”

Avoid the word “pulses” in this context. How about “factors”?

I suggest making the last sentence first person “Here we focus on resource…”

Paragraphs 3 & 4 –

Minor errors that need attention (South-facing aspect), space after a comma in citations, remove parenthetical clause about low shrubs.

Paragraph 5 –

Strike the first sentence and lead with the second sentence

I like the foundation developed in that paragraph.

Paragraph 6 –

Second sentence – avoid the use of “so” and replace with “and many plants” (remove flowers there).

Third sentence – no italics for family names (Syrphidae). Andrena is misspelled and I don’t wonder if more of the long-tongued solitary bee genera would be more likely Vaccinium pollinators (like Osmia spp., Megachile spp., or Anthophora)?

Paragraph 8 –

The authors should really just stick to canopy cover, active layer depth, soil moisture, and conspecific pollen loads in the first sentence. Perhaps a second sentence to draw the reader to those measures as estimates of the unmeasured factors (?).

I like the signpost paragraph and the predictions. Specify that by pollen load you mean conspecific pollen.

Methods:

Correct italicized parentheses.

Remove Figure 1.

Describe what constitutes a ramet

In Statistical Analyses section: correct the last sentence of the first paragraph. It would be “Values for the first two PCA axes…” not “both PCA axes” as there would be n – 1 axes in PCA analysis.

Correct font in the second paragraph in that section.

Include “Stand Age” in Table 1 and elsewhere.

Throughout the manuscript statistical symbols should be italicized.

Spaces should be placed on either side of “ = “ signs, etc.

Results:

SEM model fit section – first sentence is unclear. Do you mean to say all the “sites” included rather than “ramets”? Later on the SEMs are separated by sites. It is not very clear that each species was being treated separately. The second clause of that sentence if very difficult to interpret. I would suggest something like “… fit poorly as measured by three standard model fit metrics (…”

Limitations for flower and fruit production: canopy cover section – I would suggest striking “…and environment” in 3rd sentence.

Direct and indirect effects section – I would avoid the use of the pronoun “it” in the second sentence.

Table 2 and 3 legends. Please write out what “TFR” stands for.

Table 4 and 5 legends. Please include what statistic is being reported (correlation coefficient/Pearson’s r presumably).

Discussion:

First sentence – I would recommend striking “in this research” as it is implied and not necessary.

Pollen limitation section – I partially disagree with the second sentence. While blueberry is self-compatible it is protandrous and sets basically no fruit in the absence of pollinators (various studies).

Resource limitation section – Don’t forget that bees and many other pollinators have much lower abundance and activity levels in the canopy, for example in our Alaska bee biodiversity surveys we are catching 22 x the number of bees in open habitats relative to adjacent boreal forest – and there should be numerous studies showing this pattern outside of Alaska.

6th paragraph first sentence is incomplete.

Study limitations sections – good points in the first two paragraphs

6. PLOS authors have the option to publish the peer review history of their article (what does this mean?). If published, this will include your full peer review and any attached files.

Reviewer #1: No

Reviewer #2: Yes: Matthew L. Carlson

---

## [Author Response · Author response to Decision Letter 0]

4 Mar 2020

Point-by-point Response to the reviewers' comments 

Editorial Requirements:

 My sincerest apologies for any mistakes. 

2. In your Methods section, please provide additional location information of the sampling sites, including geographic coordinates for the data set if available.

We have added GPS coordinates to the site description table in supplementary material S1 Table. 

3. In your Methods section, please provide additional information regarding the permits you obtained for the work. Please ensure you have included the full name of the authority that approved the sampling sites access and, if no permits were required, a brief statement explaining why.

 I did not personally obtain permits. The sites are part of the Bonanza Creek Long Term Ecological Research Site and researchers of the LTER are allowed access through agreements with the State of Alaska, the Bureau of Land Management, and the Department of Defense. We added a statement to the first paragraph of the Methods section. 

(Lines 147-148) No permits were necessary for the work; LTER scientists are allowed access to the sites through agreements with the US Forest Service, State of Alaska, and the Department of Defense.

4. We note that Figure 1 in your submission contains map images which may be copyrighted. We require you to either (a) present written permission from the copyright holder to publish these figures specifically under the CC BY 4.0 license, or (b) remove the figures from your submission.

Reviewers suggested the map was unnecessary and it has been removed from the manuscript. 

Dear Reviewers, 

Our sincerest thanks for the time and attention you spent on this manuscript. Your comments helped create a clearer manuscript of more scientific merit than the original draft.

Please note that line numbers in the manuscript restart in the middle of the Results section due to an error caused by Table 2 being in landscape position. In the responses to your comments line numbers referencing the first part of the document look like (Lines ## - ##) and the second part of the document (Lines 2: ## - ##). I’m sorry for the trouble. 

Lindsey

Reviewer #1:

Reviewer #1: In the study, the relative effects of light (canopy cover), nutrient resources (depth of the active layer and soil moisture), and pollen availability (pollen load) were assessed on flower and berry production in the boreal forest around Interior Alaska. The study is a useful contribution, and a great effort was given to study a complex system of factors affecting on berry plants. The use of structural equation model (SEM) is well justified to analyse relationships among unobserved concepts using the above mentioned variables observed. Data collected in a single year only is understandable (due to high work load in the field and lab) and is not limiting the value of the study. I have comments, some of which may be regarded as pointing some shortcomings.

1. In Introduction, the authors present several assumptions without any references (“we expect”, “may “be, “may affect”, etc.). I suggest that you try to find references to support your expectations/assumptions, and separate your hypotheses from the results published in literature. For example, Finnish studies by Anne Tolvanen and Jussi Kuusipalo (1988, Vegetation 76: canopy cover and fruiting of bilberry) may support some of your speculations and results.

Thank you for pointing this out, our other reviewer had some similar concerns. Below is a sampling of the citations we have added to the introduction and have worked to more explicitly support our claims throughout the manuscript. 

(Lines 77-79) We would therefore expect fire history to affect resource availability both by altering the canopy cover and by altering soil moisture and depth of thaw as has been seen in other northern regions such as Fennoscandia [30].

(Lines 86-88) We would therefore expect a stronger relationship between canopy cover and investment in reproduction in blueberries than in lingonberries [12, 34].

(Lines 88-92) Similarly, because of their higher nutrient demands, blueberries may be more negatively impacted by low soil nutrients than lingonberries. Previous experimental work in the region found V. uliginosum showed a stronger growth response to fertilization than V. vitis-idaea, and in the natural system nitrogen concentrations in V. uliginosum are diluted throughout the season as the plants continue to grow, but the same is not true in V. vitis-idaea [35-36]

2. According to the caption of Fig. 3, the size of the site seems to be 50 m x 60 m. This should be mention already in Methods.

 Good point. We have added the information in the first Methods paragraph.

(Lines 143-145) We evaluated berry production at 17 sites, each 50 m x 60 m, within the Bonanza Creek LTER Regional Site Network in the 2017 growing season (S1)

3. In Methods (and Abstract), please mention as early as possible that the measurements took place in a single growing season in 2017.

 Added to the Abstract and the first paragraph in the Methods

(Lines 20-24) We evaluated how fruit production in two boreal shrubs, Vaccinium uliginosum (blueberry) and V. vitis-idaea (lingonberry), was explained by factors associated with resource availability (such as canopy cover and soil conditions) and pollen limitation (such as floral resources for pollinators and pollen deposition) across boreal forest sites of Interior Alaska in 2017. 

(Lines 143-145) We evaluated berry production at 17 sites, each 50m x 60m, within the Bonanza Creek LTER Regional Site Network in the 2017 growing season (S1).

4. You assessed flowers of other species flowering at the same time as Vaccinium species studied. What about species flowering earlier in the spring, and thus luring and sustaining pollinators at the site. Should you assess these species, especially willows as well?

The reviewer brings up a good point. However, blueberry is one of the first insect-pollinated species to flower in this habitat (pers. ob.; see also Mulder and Spellman 2017). There were several species of willow at some sites but many willows are primarily wind pollinated. 

(Lines 96-97) V. uliginosum is one of the first insect-pollinated species to flower in this habitat [pers. obs.; see also 38].

5. Any reason to count total floral resources using different radius for blueberry (1 m) and lingonberry (0.5 m)?

That is a mistake. We measured ramet density across the sites with different radii to save time since lingonberry ramets are smaller and grow more densely. Total floral resources was measured for both species within half a meter radius. Our apologies.

(Lines 196-200) We counted total floral resources, defined as all flowers of any species within 0.5 m radius of the focal blueberry or lingonberry ramet, during peak flowering of the Vaccinium as a measure of the potential for neighborhood plants to attract pollinators to the area or compete with focal plant flowers for resources. R. groenlandicum and C. calyculata produce many flowers per inflorescence and represent the majority of non-Vaccinium floral neighbors.

6. In PCA, you could calculate and present Cronbach’s alpha if variable deleted to explore the internal consistency of the variables included in PC analysis.

Cronbach’s alpha of all the standardized PCA values is 0.64 (0.60-0.69 is “Questionable” according to George and Mallory 2003). If middle sites BFY6 and GSM4 are excluded, alpha becomes 0.70 (“Acceptable”). Cronbach’s alpha would improve with the removal of either time-since-fire or soil temperature. While those values may help distinguish differences between the two environmental descriptions of the two habitat types, those two variables are necessary to the growth of Vaccinium and I think should stay in the PC values used later. 

(Lines 267 - 271) . Cronbach’s alpha of the two PCA groups was 0.64, “Questionable” according to George and Mallery 2003 [64]. Cronbach’s alpha improves to > 0.70 (“Acceptable”) when we remove soil temperature or time since fire but as both are thought to be important to berry production, we sacrificed a strong group distinction in the PCA for a hopefully more explanatory SEM.

7. Please use consistently “Age” or “Time since fire”. In Table 1, you use Age, and in Fig. 3, Time since fire.

Thank you for pointing out that inconsistency. We went through the paper to make sure it is “time since fire” consistently.

8. “Canopy cover explained more than twice as much of the change in allocation to leaves in lingonberry as it did in blueberry (R2 =0.24 vs. R2 =0.10).” I am not sure if you can compare two R2 values obtained for two different data sets.

You are correct. We have reworded the sentence to talk about the relative size of the effect to each species. 

(Lines 34 - 35) Canopy cover explained substantial variation of the change in allocation to leaves in lingonberry but not in blueberry (R2 = 0.24 vs. R2 = 0.10). 

9. I guess simple linear regression models (no equations were presented) were fitted to explain the biomass ratios as a function of canopy cover. You have a hierarchical data structure; ramets within sites? Therefore, you should use a multi-level modelling in fitting the regression equations.

 Ratio ~ (site mean canopy [e.g. BFY4]) + (ramet canopy - Site mean canopy)

We reran the biomass ratios in a hierarchical regression to see whether it supported the explanation in the Discussion that the positive relationship between investment in berries and canopy cover is driven by within-site variation. It doesn't: the positive relationship in lingonberry in lowland sites is driven by the site level, not the ramet level. We have added explanations in the discussion. 

(Lines 84 - 90) The positive correlation between canopy and berry production in lowland sites for both species may be the result of relatively less investment in leaves, [CM1] [LP2] leading to higher productivity in both Vaccinium and its neighboring species. This is supported by the higher blueberry fruit set (the ratio of berries to flowers) with higher canopy cover at low elevation sites and hierarchical regressions in which the biomass investments of lingonberry ramets were significantly related to mean canopy cover of the site level but not the canopy cover immediately above the ramet.

10. The number of observations is missing in Tables 4 and 5. In Fig 7, the number of observations is 15. Why? The number of sites was 17.

We added sample size to tables 4 and 5.

Table 4: Blueberry, n = 186; lingonberry, n = 205.

Table 5: Upland blueberry, n = 80; lowland blueberry, n = 106; upland lingonberry, n = 97; lowland lingonberry, n = 98.

For the measurements in Figure 7 (now Figure 5) we needed both canopy cover and reproductive rate of the site’s Vaccinium. We were only able to measure canopy cover at 16 sites. We had ATV troubles at the time of data collection so a remote site had to be left out. Then, in our random sample of ramet density, site BFY6 had no flowering lingonberry ramets and site WDI6 had no flowering blueberry ramets. The 1-2 ramets included in the study from those sites were from an exhaustive search of the site after our attempts at random sampling failed to hit any. 

11. The authors discuss possible impacts of the expected changes in boreal forest fire regime, vegetation composition and soils on berry productivity. I am not aware of forest management in Alaska. If managed, then the effect of forest management (affecting the canopy cover) on berry production should also be discussed.

There is occasional logging, or fire suppression when a wildfire is near human infrastructure, but the area is primarily unmanaged. 

Reviewer #2: Dr. Matthew Carlson

General:

This study explores how site variation relates to flower and fruit production in two species of Vaccinium in Alaska. Specifically the study tests predictions of patterns in abiotic resource and pollen limitation in fruit production related to time since last fire, total local floral resources, and differences in the two species’ life histories. The primary results are analyzed using structural equation modeling that offer insights into direct and indirect effects on flower and fruit production and differences between upland and lowland habitats. Overall the results are in accordance with expectations, although the authors found evidence for total floral resources affecting pollen loads or the number of berries in only one of four contexts.

In general, the underlying questions are of ecological interest, the methods and the analyses appear sound to my knowledge (though I have one or two questions), and the interpretation of results is appropriate. The grounding of research in fruit limitation research could be strengthened. The writing, formatting, and figures could be substantially improved. Overall, I believe it is a fine study and sufficient manuscript to warrant publication with some improvements and questions addressed. I provide some general statements and questions followed by specific edits/suggestions below.

I recommend the authors develop a stronger foundation of the study species’ reproductive biology and pollen limitation on fruit production in high latitude contexts. Interpretation of their results would be improved information by providing context to highly relevant studies that are not referenced – e.g., Jacquemart & Thompson 1996; Jacquemart 1997; Brown & McNeil 2009, a number of Totland’s papers; Fulkerson et al. 2012; Urbanowicz et al. 2018. For example, V. uliginosum is self-compatible where it has been studied, but does not auto-pollinate in the absence of visitors. Often open pollination treatments result in equivalent proportions of fruit set (but not seed set) as supplemental hand pollinations. Pollen limitation and resource limitation of reproduction are typically measured as a proportion of fruits to flowers or seeds to flowers (or ovules) under experimentally enhanced and control conditions - there is a value in the sense that the authors use the terms and by using the natural environmental variation, but referencing the distinction would be useful.

(Lines 103 - 108) Pollen availability explained the most variation in Finnish bilberry (V. myrtillus) fruit production models [43] and was a limiting factor in fruit set of V. uliginosum in Greenland as well [44]. Between the two focal species, V. vitis-idaea flower structure is more adapted to cross-pollination than V. uliginosum [45]. In experiments, cross-pollination led to more fruits than self-pollination in V vitis-idaea but V. uliginosum had similar levels of fruit production regardless of whether cross- or self-pollinated [46-48].

I would recommend that the authors acknowledge pollen limitation may be a function of both pollen quantity and pollen “quality” (i.e., amount of outcrossed pollen). Pollen quality limitation can be an issue especially for highly clonal plants even with relatively abundant pollinators. Vaccinium vitis-idaea is self-incompatible and therefore more likely to be subject to geitonogamous pollinations that don’t result in fruits or seeds.

While I concur with the general relationship between active layer depth, soil moisture, and nutrient availability, however that relationship is noisy and can be variable within sites. Thus we are presented with a proxy for soil nutrient availability with unclear relationship to what the plants actually have access to. The authors recognize this and briefly discuss it in the Methods, which is appropriate. If tissue from the ramets has been retained following the biomass measurements, could leaf N content or concentration at least be measured at least? Has anyone else measured soil N and P at these LTER sites? Obviously it is too late in the year to get soil samples now.

We wanted to include soil nutrient availability but it wasn’t something we had the budget for and unfortunately is not a measurement that has been collected from the LTER regional site network with any sort of controlled process. 

Some more description of what the authors consider a ramet for both of these species would be warranted in the Methods, as what constitutes the “ground” in boreal forests with deep layers of loose organic layers or moss can be ambiguous, especially coupled with the extensive branching of these species low to the ground.

(Lines 152 - 153) We defined an individual, hereafter a ramet, as a single aboveground stem that did not branch within 2 cm of the soil or moss surface. 

It should be stated in the pollen availability section that the authors were only counting conspecific tetras or no heterospecific tetrads were observed.

(Lines 204 - 205) We estimated pollen availability by collecting two pistils from conspecific flowers near each study ramet and estimating conspecific pollen loads on the stigmas under a microscope. 

(Lines 209 - 210) Following Spellman et al. 2015, a ramet was considered “well-pollinated” when the mean number of conspecific pollen tetrads on neighboring stigmas was >10. 

The authors should note throughout the manuscript that canopy cover/light availability not only may affect flower and fruit production through physiological factors, but also due to increased pollinator activity – there should be plenty of references in the literature.

 We added a few mentions, particularly in the discussion

(Lines 112 - 115) Environmental conditions can also affect pollinator activity. In general, pollinators are expected to be more active in warmer sites; bees are strongly limited by temperature in Interior Alaska [50]. Bumblebees are less affected by temperature but, in Interior Alaska, solitary bees are at their lowest abundance in closed forests [51].

(Lines 252 - 254) Canopy cover was expected to affect flower and fruit numbers directly through light availability but also indirectly through pollinator activity and by acting as a proxy for local growing conditions.

It would make more sense in my mind to drop the two sites (GSM4 and BFY6) that were intermediate in environmental characteristics as described by the PCA rather than assigning them a lowland category. Or I suppose a hierarchical cluster analysis or other analysis that assigns groups could be a more defensible approach for determining where the two sites should go.

Reviewer 1 had similar concerns and suggested we use Cronbach’s alpha to verify differences between the groups. Cronbach’s alpha of the standardized PCA values is 0.64 (0.60-0.69 is “Questionable” according to George and Mallery 2003). If middle sites BFY6 and GSM4 are excluded, alpha becomes 0.70 (“Acceptable”). However, the SEM without those observations does not adhere to the three fit statistics. Cronbach’s alpha would also improve with the removal of either time-since-fire or soil temperature. While those values may help distinguish differences between the environmental descriptions of the two habitat types, those two variables are necessary to the growth of Vaccinium and I think should stay in the PC values used later. 

The SEM has some relationships that seem to me to be lacking and potentially important. Why would there not be estimation of the effect of # Flowers on Pollen Load? Only Total Floral Resources on Pollen Load? It is likely of much less importance how many Rhododendron and other flowers are present on pollen loads than number of Vaccinium flowers. What about the relationship between canopy cover and total floral resources?

We are limited in the number of connections we can make within the model. Given our sample size we are already on the high end of the acceptable ratio of connections : sample size. Total Floral Resources does include conspecific flowers in the area. . 

There are a lot of SEM figures and the first model could be included as a figure only in the supplemental figures section. Removing it from the primary manuscript would improve the clarity of the message.

 Okay. It is now in the supplemental (S2). 

There are some sections of the manuscript that read smoothly and the overall architecture of paragraph succession is logical, but much of it is packed with unnecessary parenthetical clauses, missing punctuation, sentences that could be improved for clarity, and poor overall formatting. I would encourage the authors to spend a bit more time and carefully craft the language and formatting. I will include a few suggestions below in the specific section, but the number of edits would be too lengthy to include all corrections or suggestions.

Specific:

We sincerely appreciate Dr. Carlson’s specific reviews on grammar and punctuation. As it didn’t seem necessary to craft a specific response to every edit we have sometimes added the lines where we made the suggested changes without adding a comment. Thank you for your understanding. 

Figure 1: I would remove this figure. You include a supplemental table that could include latitude longitudes as well. If you do feel it is necessary it could use some cartographic/design attention (both the inset and the primary maps are very difficult to discern features and the figure legend does not do an adequate job of explaining what the reader is looking at.

We removed Figure 1 from the manuscript and included lat-long in supplemental table S1. 

Figure 4: Please remove the raw data points. The inclusion of the points makes the figures unnecessarily busy and distracting. It is not clear from the axis label or legend what “Pollen Load” is (pollen grains/stigma or pollen tetrads/stigma… not pollen grains/ramet as suggested by the legend?). Also I would suggest the y-axis label of # Berries, say Berries per ramet.

 Thank you for the suggestion. The figure should now reflect all the suggested changes. 

Title: “Spatial variability…” is misleading since there is a whole subdiscipline of ecology that explores spatial patterns and variability and this study does not explicitly address space. Something along the lines of: “Patterns of pollen and resource limitation of fruit production in Vaccinium uliginosum and V. vitis-idaea in interior Alaska” or try and squeeze in some primary variables: “Pollen and resource limitation of fruit production in Vaccinium uliginosum and V. vitis-idaea in interior Alaska: effects of landscape position, stand age, and canopy cover on reproduction”

Both good suggestions. We changed the title to reflect the first suggestion and removed the two mentions of “spatial variability” from the text.

Abstract: Looks okay other than far too many parentheses and in the last sentence it is really pollination service or pollinator service, not the raw abundance of pollinators that is important in fruit production.

We removed 4 out of 6 parentheticals. I kept the two others because I like listing the specific variables we used. 

Introduction:

Paragraph 1 -

Missing a period in the first sentence.

(Line 35) At least 50 species of plants produce fleshy fruits (hereafter: “berries”) in Alaska [1].

Avoid the long list of common names – the single more often used one would be appropriate.

(Lines 37 - 39) Vaccinium vitis-idaea L. (lingonberry) and V. uliginosum L. (lowbush blueberry, hereafter: blueberry) are two of the fruits most commonly consumed by both humans and animals [2]. 

“Ursus spp.” should be “Ursus spp.” or just say Ursus arctos and U. americanus. There are only two after all.

(Lines 44 - 45) Many species including bears (Ursus arctos and U. americanus), foxes (Vulpes vulpes), and voles (e.g., Myodes rutilus) eat the berries [3-6].

Branta hutchinsii reference to Hupp’s paper – those are a more coastal – nesting species that I believe he was emphasizing feed on Empetrum. Maybe I’m wrong and he did talk about them feeding on Vaccinium.

(Lines 39 - 40) Many species including bears (Ursus arctos and U. americanus), foxes (Vulpes vulpes), and voles (e.g., Myodes rutilus) eat the berries [3-6].

Paragraph 2 –

“…pollinator availability” I would recommend changing to “pollinator service (limiting pollen compatible deposition and subsequent fertilization)”

(Lines 49 - 53) Understanding how Vaccinium berry production responds to heterogeneous environmental factors such as variation in resource availability (limiting growth and berry development) and variation in pollinator service (limiting pollen compatible deposition and subsequent fertilization) within Alaska’s boreal forest can provide a foundation for modelling berry crops for humans and animals. 

I would strike (i.e. where on the landscape we would expect changes in berry production)

(Lines 49 - 53) Understanding how Vaccinium berry production responds to heterogeneous environmental factors such as variation in resource availability (limiting growth and berry development) and variation in pollinator service (limiting pollen compatible deposition and subsequent fertilization) within Alaska’s boreal forest can provide a foundation for modelling berry crops for humans and animals.

“…growth and fruit availability are not always correlated” I would say “… growth, fruit, and seed production are not always correlated with one another [21].”

(Lines 55 - 56) However, vegetative plant growth, fruit, and seed availability are not always correlated with one another [20]. 

“Fruit production in all plants…” should read “Fruit production in all flowering plants…”

(Lines 56 - 58) Fruit production in all flowering plants is limited by four factors: 1) resources (e.g., light and soil moisture), 2) pollination, 3) external factors such as herbivory, disease, or harsh weather, and 4) genetics [21-23].

Avoid the word “pulses” in this context. How about “factors”?

(Lines 56 - 58) Fruit production in all flowering plants is limited by four factors: 1) resources (e.g., light and soil moisture), 2) pollination, 3) external factors such as herbivory, disease, or harsh weather, and 4) genetics [21-23].

I suggest making the last sentence first person “Here we focus on resource…”

 (Lines 58 - 59) Here we focus on resource and pollen limitation. 

Paragraphs 3 & 4 –

Minor errors that need attention (South-facing aspect), space after a comma in citations, remove parenthetical clause about low shrubs.

 Thank you for pointing out these errors. 

Paragraph 5 –

Strike the first sentence and lead with the second sentence

 We made the suggested change. (Line 80)

I like the foundation developed in that paragraph.

 Thank you!

Paragraph 6 –

Second sentence – avoid the use of “so” and replace with “and many plants” (remove flowers there).

(Lines 94 - 95) Pollinator and floral diversity are low in the boreal forest, many plants use multiple pollinator species, and those pollinators visit many flower species [37]. 

Third sentence – no italics for family names (Syrphidae). Andrena is misspelled and I don’t wonder if more of the long-tongued solitary bee genera would be more likely Vaccinium pollinators (like Osmia spp., Megachile spp., or Anthophora)?

Those solitary bees are confirmed pollinators in many other areas where Vaccinium grows, particularly if there is commercial growing, but I don’t see those species mentioned in the few papers from Alaska. I did add Lasioglossum spp. 

(Lines 97 - 100) In Interior Alaska, bumblebees (Bombus spp.), syrphid flies (Syrphidae), and solitary bees (e.g., Andrena spp., and Lasioglossum spp.) carry the most blueberry and lingonberry pollen [39-40]. Bee genera that are present in Interior Alaska and are known to pollinate Vaccinium in other regions include Osmia spp., Megachile spp., and Anthophora [41]. 

Paragraph 8 –

The authors should really just stick to canopy cover, active layer depth, soil moisture, and conspecific pollen loads in the first sentence. Perhaps a second sentence to draw the reader to those measures as estimates of the unmeasured factors (?).

Removed some of the parentheses. I think discussion later in the paper can cover the fact that measures were a stand in for nutrients 

(Lines 118 - 120) We assessed the relative effects of light (as indicated by canopy cover), depth of the active layer, soil moisture, and conspecific pollen load on flower and berry production in the boreal forest around Interior Alaska. 

I like the signpost paragraph and the predictions. Specify that by pollen load you mean conspecific pollen.

(Lines 118 -120) We assessed the relative effects of light (as indicated by canopy cover), depth of the active layer, soil moisture, and conspecific pollen load on flower and berry production in the boreal forest around Interior Alaska. 

Methods:

Correct italicized parentheses.

Ok.

Remove Figure 1.

 We have removed Figure 1 and instead have lat-long included in Supplemental Table S1

Describe what constitutes a ramet

(Lines 152 - 153) We defined an individual, hereafter a ramet, as a single aboveground stem that did not branch within 2 cm of the soil or moss surface. 

In Statistical Analyses section: correct the last sentence of the first paragraph. It would be “Values for the first two PCA axes…” not “both PCA axes” as there would be n – 1 axes in PCA analysis.

(Lines 230 - 231) Values for the first two PCA axes were used as explanatory variables in the structural equation models. 

Correct font in the second paragraph in that section.

 Thank you for pointing it out. 

Include “Stand Age” in Table 1 and elsewhere.

Reviewer 1 pointed out we were using “time since fire” and “stand age” interchangeably. We have changed the manuscript to “time since fire” consistently.

Throughout the manuscript statistical symbols should be italicized.

 Thank you for pointing out that pervasive mistake. I believe we have now found them all. 

Spaces should be placed on either side of “ = “ signs, etc.

 Thank you for pointing that out. 

Results:

SEM model fit section – first sentence is unclear. Do you mean to say all the “sites” included rather than “ramets”? Later on the SEMs are separated by sites. It is not very clear that each species was being treated separately. The second clause of that sentence if very difficult to interpret. I would suggest something like “… fit poorly as measured by three standard model fit metrics (…”

 Removed some pieces of that section and thank you for the wording suggestion. 

(Lines 319 - 325) Our first multi-group SEM, including all sites but divided by species, fit poorly with none of the three metrics falling in the proper range (CMIN/df of 4.290 [good fit: 1-5], CFI was 0.806 [fit: >0.90], and RMSEA was 0.093 [90%CI: 0.070 –0.118; fit: 0.05 inclusive])(S2). The models using all sites explained 31% of the variation in blueberry fruit production but only 9% of lingonberry fruit production. When models were run after separating the data by upland and lowland sites, fit statistics improved: CMIN/df was 2.735, CFI was 0.902 and RMSEA was 0.068 (90% CI: 0.050 - 0.086); more paths were significant, and R2 values improved (Fig 4).

Limitations for flower and fruit production: canopy cover section – I would suggest striking “…and environment” in 3rd sentence.

(Lines 352 - 354) When upland and lowland sites were evaluated separately, canopy cover had differing effects on flower production depending on the species (Fig 4). 

Direct and indirect effects section – I would avoid the use of the pronoun “it” in the second sentence.

(Line 366 - 368) However, stand history had indirect effects as well: stand history was strongly positively related to canopy cover...

Table 2 and 3 legends. Please write out what “TFR” stands for.

In my copy “‘TFR’ is total floral resources” is included in the caption under the table. Perhaps I loaded something wrong to the PLOS ONE site. 

Table 4 and 5 legends. Please include what statistic is being reported (correlation coefficient/Pearson’s r presumably).

e.g. Table 5 Parameter estimate (correlation coefficient) and p values for all relationships. Significant (p < 0.05) relationships are bold and contain the adjusted R2 value. Upland blueberry, n = 80; lowland blueberry, n = 106; upland lingonberry, n = 97; lowland lingonberry, n = 98.

Discussion:

First sentence – I would recommend striking “in this research” as it is implied and not necessary.

(Lines 2: 49 - 51) Our primary goal was to assess pollen versus resource (light and nutrient) limitation on berry production of blueberry and lingonberry across the landscape in black spruce of Interior Alaska. 

Pollen limitation section – I partially disagree with the second sentence. While blueberry is self-compatible it is protandrous and sets basically no fruit in the absence of pollinators (various studies).

We added more about Vaccinium reproduction to both the introduction and the discussion

(Lines 103 - 108) Pollen availability explained the most variation in Finnish bilberry (V. myrtillus) fruit production models [43] and was a limiting factor in fruit set of V. uliginosum in Greenland as well [44]. Between the two focal species, V. vitis-idaea flower structure is more adapted to cross-pollination than V. uliginosum [45]. In experiments, cross-pollination led to more fruits than self-pollination in V vitis-idaea but V. uliginosum had similar levels of fruit production regardless of whether cross- or self-pollinated [46-48].

(Lines 2: 60 - 65) Lingonberries were more pollen limited than blueberries, especially in lowland sites (Table 3). Lingonberry is partially self-incompatible [48], may be more dependent on pollinators for fertilization and suffer from geitonogamous pollination more than blueberries[CM1] . In experiments, V. uliginosum produces the same number of fruits whether the experimenters self-pollinated or cross-pollinated the plants. However, V. uliginosum will not self-pollinate in the absence of visiting pollinators [45].

Thank you for sharing the Urbanowicz study in your opening statement. It was a good read and one I had missed in my literature review. 

Resource limitation section – Don’t forget that bees and many other pollinators have much lower abundance and activity levels in the canopy, for example in our Alaska bee biodiversity surveys we are catching 22 x the number of bees in open habitats relative to adjacent boreal forest – and there should be numerous studies showing this pattern outside of Alaska.

We worked to clarify this point in both the introduction and discussion sections. 

(Lines 112 - 117) In general, pollinators are expected to be more active in warmer sites; bees are strongly limited by temperature in Interior Alaska [50]. [LP1] Bumblebees are less affected by temperature but, in Interior Alaska, solitary bees are at their lowest abundance in closed forests [51]. However, high canopy cover may also be indicative of good growing conditions for deciduous species, and result in high floral resources, resulting in a complex relationship between canopy cover and pollinator activity.

(Lines 2: 100 - 102) Though we didn’t directly measure pollinators, many other studies have also found higher canopy cover leads to lower abundance and activity levels of pollinators, leading to an indirect effect on pollen limitation [51].

6th paragraph first sentence is incomplete.

 Thank you.

Study limitations sections – good points in the first two paragraphs

 Thank you.

---

## [Decision Letter · Decision Letter 1]

22 Jul 2020

Patterns of pollen and resource limitation of fruit production in *Vaccinium uliginosum* and *V. vitis-idaea* in Interior Alaska

PONE-D-19-27691R1

Dear Dr. Parkinson,

We’re pleased to inform you that your manuscript has been judged scientifically suitable for publication and will be formally accepted for publication once it meets all outstanding technical requirements.

Kind regards,

Harald Auge

Academic Editor

PLOS ONE

Additional Editor Comments (optional):

Reviewers' comments:

Reviewer's Responses to Questions

**Comments to the Author**

1. If the authors have adequately addressed your comments raised in a previous round of review and you feel that this manuscript is now acceptable for publication, you may indicate that here to bypass the “Comments to the Author” section, enter your conflict of interest statement in the “Confidential to Editor” section, and submit your "Accept" recommendation.

Reviewer #1: All comments have been addressed

2. Is the manuscript technically sound, and do the data support the conclusions?

Reviewer #1: Yes

3. Has the statistical analysis been performed appropriately and rigorously? 

Reviewer #1: Yes

4. Have the authors made all data underlying the findings in their manuscript fully available?

Reviewer #1: Yes

5. Is the manuscript presented in an intelligible fashion and written in standard English?

Reviewer #1: Yes

6. Review Comments to the Author

Reviewer #1: The authors have addressed the comments on the previous version of the manuscript so that this manuscript is now acceptable for publication.

7. PLOS authors have the option to publish the peer review history of their article (what does this mean?). If published, this will include your full peer review and any attached files.

Reviewer #1: No

---

## [Editor Report · Acceptance letter]

7 Aug 2020

PONE-D-19-27691R1 

Patterns of pollen and resource limitation of fruit production in *Vaccinium uliginosum* and *V. vitis-idaea* in Interior Alaska 

Dear Dr. Parkinson:

I'm pleased to inform you that your manuscript has been deemed suitable for publication in PLOS ONE. Congratulations! Your manuscript is now with our production department. 

Kind regards, 

on behalf of

Dr. Harald Auge 

Academic Editor

PLOS ONE